# Performance Improvement of DTC-SVM of PMSM with Compensation for the Dead Time Effect and Power Switch Loss Based on Extended Kalman Filter

**Doo-Il Son, Jun-Seo Han, Je-Suk Park, Hee-Sun Lim**  **and Geun-Ho Lee ***

Electric Motor Control Laboratory, Graduate School of Automotive Engineering, Kookmin University, Seoul 02707, Republic of Korea
* Correspondence: motor@kookmin.ac.kr; Tel.: +82-2-910-6409

**Abstract:** Two algorithms have been extensively studied for motor control: Field Oriented Control (FOC) and Direct Torque Control (DTC). Both control algorithms use a Voltage Source Inverter (VSI) to drive a Permanent Magnet Synchronous Motor (PMSM). To prevent short-arm short-circuit accidents when driving PMSM using VSI, a dead time is used to turn off the TOP and BOTTOM switches of each arm at the same time. However, this dead-time technique causes an unexpected pole voltage to be applied to the PMSM on the VSI output voltage, causing distortion and resulting in control nonlinearity. The disturbance voltage that causes nonlinearity is difficult to measure directly with the sensor. Therefore, this paper analyzes the nonlinearity of the controller due to the distorted voltage caused by the dead time during PMSM operation using the DTC algorithm and predicts the distorted output voltage using the extended Kalman Filter (EKF) to improve control stability. As a result, The algorithm proposed in this paper has verified the improvement of torque ripple and stator flux ripple through experiments and simulations.

**Keywords:** PMSM; direct torque control; Kalman filter; dead-time; IGBT; inverter; FOC; DTC-SVM

## 1. Introduction

In the field of motor control, FOC (field-oriented control) and DTC (direct-torque control) have been predominantly studied. The DTC algorithm has been in development since the 1980s. The DTC algorithm was widely used in the field of induction motors in their early days. As high efficiency and high density became important in the motor industry, the DTC algorithm also began to be applied to the synchronous motor field. The DTC algorithm is a control method that follows the torque and magnetic flux of the motor using a hysteresis controller and a lookup table (LUT) for the torque command value and stator flux command value [1–3]. However, the conventional DTC algorithm using a hysteresis controller and LUT has a significant disadvantage in that torque ripple occurs because the load angle between the rotor flux and the stator flux cannot be maintained constantly during motor rotation. To overcome this, many control schemes have been proposed. Among them, a DTC-SVM (Direct Torque Control Space Vector Modulation) technique that is able to instantaneously keep the load angle constant according to the position of the stator flux has been developed [4–9]. In this way, a DTC-SVM control technique that places the stator flux at an accurate location has been proposed to reduce torque ripple, but additionally, for accurate DTC-SVM control, the position and magnitude of the stator flux must be accurately estimated. Most DTC-SVM algorithms use motor parameters and current in the method of estimating the stator flux and position. Therefore, the magnitude of the stator flux is estimated using the three-phase current value of the motor measured using a current sensor mounted on the VSI (Voltage Source Inverter). If the noise is included and measured, the exact location and value of the stator flux cannot be estimated, causing torque ripple or poor control characteristics [10–13]. In this case, the

dead time used to prevent a short circuit when turning on/off the power switch element of the VSI is a representative disturbance, and many methods to reduce this disturbance have been proposed [14–20]. Most of these methods compensate for the dead time disturbance that occurs according to the current direction when using the FOC algorithm. In addition, regarding the corresponding compensation method, the disturbance voltage generated by the diode of the power switch device, such as the MOSFET and IGBT of the inverter, was not considered. In a later study, the diode voltage drop value of the power switch device was also considered, but the corresponding value could not be measured in real-time [20]. Therefore, this paper analyzes the effect of dead-time disturbance voltage on DTC control and uses an EKF (extended Kalman filter) to estimate and compensate for the dead-time disturbance voltage value that is difficult to measure. The Kalman filter was first applied to synchronous motors in 1983, and the algorithm has been continuously researched in subsequent decades [21]. Continuous research has been conducted in various algorithm fields of PMSMs (Permanent Magnet Synchronous Motors) using the Kalman filter. [22–25] In addition, dead-time compensation methods using Kalman filters have been continuously studied [26–30]. Most studies have focused on dead-time compensation based on the FOC algorithm. In this paper analysis of the dead-time voltage distortion that occurs when the DTC algorithm is used, and the paper proposes a method to compensate for it using an extended Kalman Filter. The effectiveness of the algorithm is demonstrated by comparing the performance of the proposed algorithm and the conventional algorithm. In addition, the voltage compensation of the proposed algorithm has been proven to improve the performance of the controller through experiments and simulations.

## 2. Direct Torque Control of PMSM

### 2.1. Mathematical Model of PMSM

The dynamic characteristics of the DTC algorithm are analyzed and the PMSM mathematical model projected to the rotor flux reference coordinate system for application to control is expressed in Equation (1). In addition, the *d*- and *q*-axis stator flux linkage equation is expressed in Equation (2), and the PMSM torque equation is expressed in Equation (3).

$$\begin{bmatrix} V_d \\ V_q \end{bmatrix} = \begin{bmatrix} R_s + \frac{d}{dt}L_d & -w_r L_q \\ w_r L_d & R_s + \frac{d}{dt}L_q \end{bmatrix} \begin{bmatrix} i_d \\ i_q \end{bmatrix} + \begin{bmatrix} 0 \\ w_r \phi_f \end{bmatrix} \tag{1}$$

$$\begin{bmatrix} \phi_d \\ \phi_q \end{bmatrix} = \begin{bmatrix} L_d & 0 \\ 0 & L_q \end{bmatrix} \begin{bmatrix} i_d \\ i_q \end{bmatrix} + \begin{bmatrix} \phi_f \\ 0 \end{bmatrix} \tag{2}$$

$$\phi'_d = L_d i_d$$

$$\begin{aligned} T_e &= \tfrac{3}{2} P(\phi_d i_q - \phi_q i_d) \\ &= \tfrac{3}{2} P(\phi_f i_q + (L_d - L_q) i_d i_q) \end{aligned} \tag{3}$$

where $V_d$ and $V_q$ represent stator voltage values of the *d*- and *q*-axis in the rotor coordinate system, respectively. $i_d$, $i_q$, $\phi_d$, $\phi_q$, $L_d$, $L_q$, $\phi_f$, $R_s$ represent the *d*- and *q*-axis stator current, the *d*- and *q*-axis stator flux, the *d*- and *q*-axis stator inductance, rotor flux, and stator resistance, respectively. If the above torque, obtained by Equation (3), is arranged in terms of the torque load angle, it can be expressed in Equation (4)

$$T_e = \frac{3}{2} P \frac{\phi_s}{L_d L_q} (\phi_f L_q \sin \delta + \frac{1}{2} \phi_s (L_d - L_q) \sin 2\delta) \tag{4}$$

Through the torque obtained in Equation (4), it can be seen that, if the stator flux linkage $\phi_s$ is kept constant, the torque can be controlled through the torque load angle $\delta$. Since the electrical response is very short compared to the mechanical response time, the load angle can be adjusted by instantaneously adjusting the position of the stator magnetic flux using the stator voltage vector.

### 2.2. Conventional DTC of PMSM

In order to apply the PMSM mathematical model to the DTC control technique, Equations (1) and (2) can be discretized with a fixed control period $T_s$ and arranged in Equation (5). In addition, if this is arranged based on the stator flux linkage standard, it can be expressed as Equation (6). As the speed increases, the magnitude of the stator resistance is relatively very small, so if ignored, it can be seen that the position of the stator flux at the point of $(k + 1)$ can be adjusted according to the $d$-axis voltage and $q$-axis voltage command through the corresponding equation.

$$V_d(k+1) = \frac{\phi_d(k+1)-\phi_d(k)}{T_s} - w_r L_q I_q(k)$$

$$V_q(k+1) = \frac{\phi_q(k+1)-\phi_q(k)}{T_s} + w_r(L_d I_d(k) + \phi_f) \tag{5}$$

$$\phi_d(k+1) = \left[V_d(k+1) + w_r L_q I_q(k+1)\right] Ts + \phi_d(k)$$

$$\phi_q(k+1) = \left[V_q(k+1) - w_r(L_d I_d(k) + \phi_f)\right] T_s + \phi_q(k) \tag{6}$$

In the case of the existing DTC algorithm, when the torque command value and stator flux command value are input into the controller using these characteristics, the state values of torque and stator flux are output through the hysteresis controller, and according to the state values, the LUT (lookup table) outputs the specified voltage command. Table 1 displays the LUTs. In most existing DTC algorithms, the torque hysteresis controller is composed of three levels and the stator flux hysteresis controller is composed of two levels. The configuration of the conventional DTC controller and the stator flux trajectory according to the voltage command vector of the conventional DTC controller are shown in Figures 1 and 2. As can be seen in Figure 2, in the case of the conventional DTC controller, when a certain torque is output, the stator flux is not located in an accurate position, resulting in poor control dynamics.

**Table 1.** LUT (lookup table) used for conventional DTC control.

| Flux | Torque | Stator Flux Section | | | | | |
|------|--------|------------|------------|------------|------------|------------|------------|
| | | $\theta_{s1}$ | $\theta_{s2}$ | $\theta_{s3}$ | $\theta_{s4}$ | $\theta_{s5}$ | $\theta_{s6}$ |
| | 1 | $V_2$ | $V_3$ | $V_4$ | $V_5$ | $V_6$ | $V_1$ |
| 1 | 0 | $V_7$ | $V_0$ | $V_7$ | $V_0$ | $V_7$ | $V_0$ |
| | −1 | $V_6$ | $V_1$ | $V_2$ | $V_3$ | $V_4$ | $V_5$ |
| | 1 | $V_3$ | $V_4$ | $V_5$ | $V_6$ | $V_1$ | $V_2$ |
| −1 | 0 | $V_0$ | $V_7$ | $V_0$ | $V_7$ | $V_0$ | $V_7$ |
| | −1 | $V_5$ | $V_6$ | $V_1$ | $V_2$ | $V_3$ | $V_4$ |

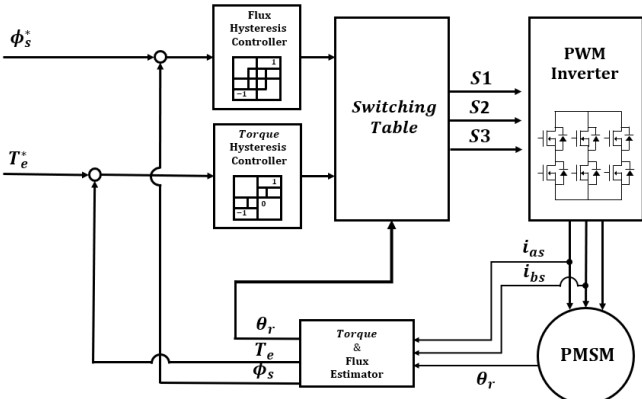

**Figure 1.** Block diagram of Conventional DTC control technique.

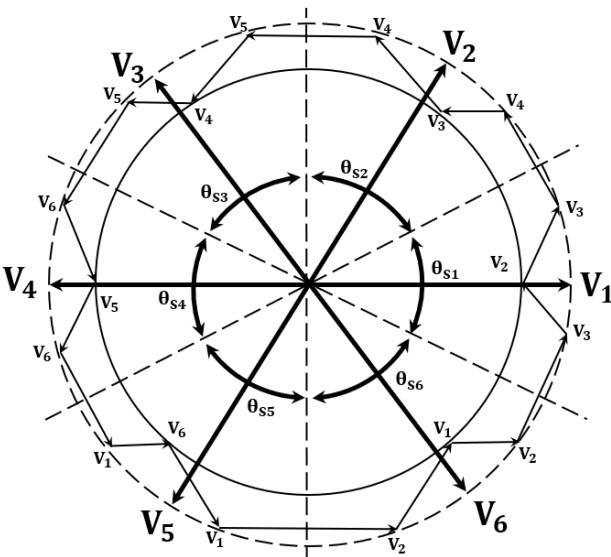

**Figure 2.** Stator flux trajectory according to the voltage vector command under conventional DTC using LUT (lookup table).

### 2.3. DTC-SVM of PMSM

The DTC-SVM control method is used to reduce the torque ripple caused by not locating the stator flux in an accurate position in the conventional DTC algorithm. The torque command value and stator flux command value are input to the controller in the same way as in the conventional DTC algorithm. However, the DTC-SVM controller outputs the torque error value as the load angle command $\delta^*$ using the PI controller, not the hysteresis controller. In addition, in the case of the stator flux controller, the angle command of the voltage command is output, through which the stator flux value can be output at an accurate position through the current motor rotation speed and the current stator flux estimation value. It outputs the stator voltage vector command value and angle command value to SVM using the load angle command and voltage command angle command output from the stator flux controller. The diagram for the DTC-SVM control block and the stator flux position difference generated when using the conventional DTC algorithm and when using the DTC-SVM algorithm are shown in Figures 3 and 4. Regarding the conventional DTC algorithm in Figure 4a, it can be seen that the stator flux vector generated using the LUT does not reach the stator flux vector $\overrightarrow{OC}$ where the load angle $\delta$ can be kept constant. This causes an error value when estimating the stator flux $\phi_s^* err$, causing torque ripple. In Figure 4b, in the case of the DTV-SVM algorithm, the state of the six power switch devices can be changed within one cycle to position the stator flux at a position $\overrightarrow{OB}$, where the load angle $\delta$ can be maintained in a constant state.

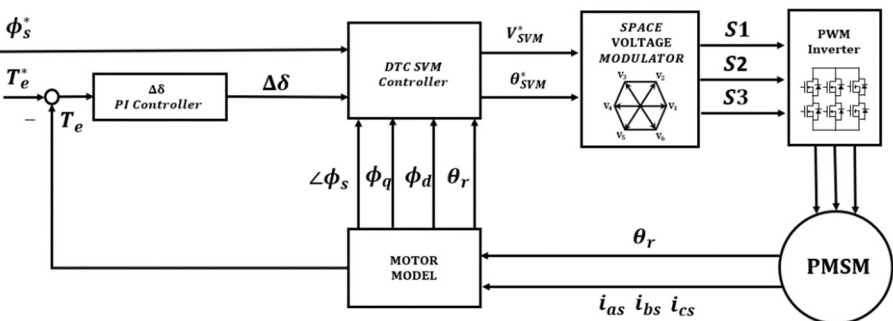

**Figure 3.** Block diagram of DTC-SVM control.

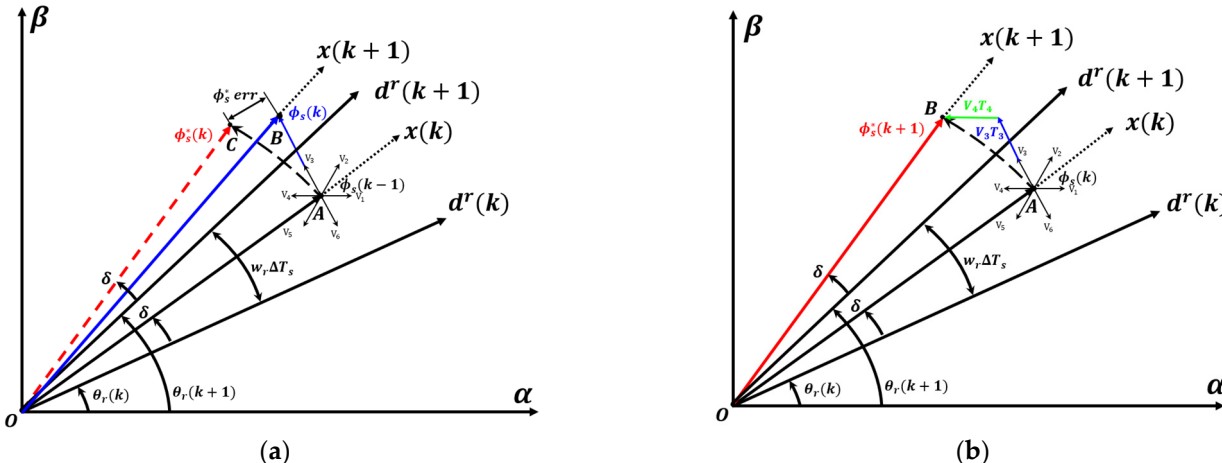

(a)                     (b)

**Figure 4.** Stator linkage flux vector trajectory according to voltage vector command under DTC-SVM control (**a**) Position of stator flux in the case of conventional DTC control (**b**) Position of stator flux in the case of DTC-SVM control.

## 3. Problems Caused by Dead-Time and Power-Switch Losses in DTC-SVM

Dead time is the time taken to turn off the top/bottom switch to prevent burnout due to a short arm when turning the top/bottom switch of each leg on or off in order to apply a pole voltage to each phase of the motor in the VSI. The overall configuration of the VSI is shown in Figure 5a, and a single leg is shown in Figure 5b. The dead time is shown in Figure 6a,b.

Turning the switch on or off during each phase applies the pole voltage output command of the controller to the motor. In the dead time, unexpected pole voltage is applied to the motor by conducting the diode of the power switch device according to the direction of the current. This is shown in Figure 7. Additionally, when the diode conducts, the loss of the forward voltage, which the diode itself has, is reflected and applied to the motor. While this is relatively small in size and can be ignored in the high-speed operation range, it is difficult to ignore in the low-speed range. In addition, the application of such unexpected disturbance voltage eventually causes a distortion of the phase current and torque ripple in the DTC-SVM algorithm, which estimates stator flux using the current value, thereby degrading performance.

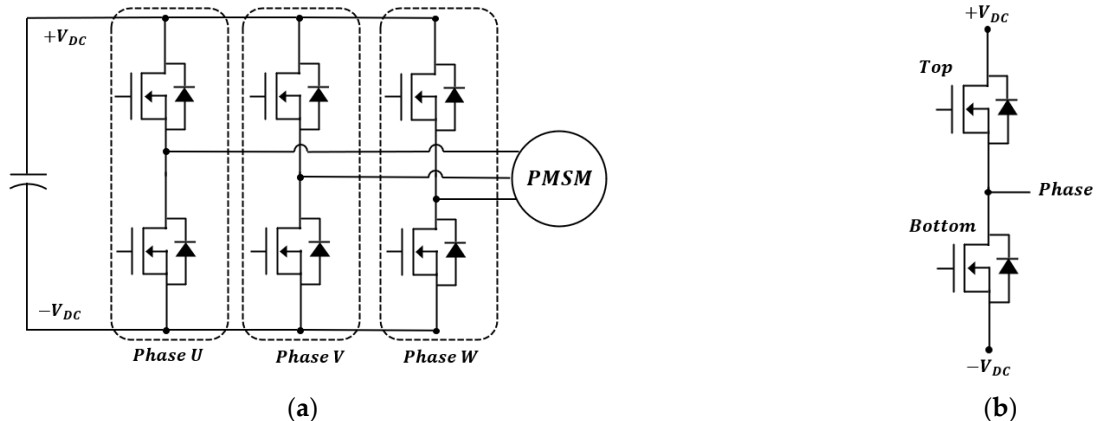

(a)                     (b)

**Figure 5.** (**a**) VSI (voltage source inverter) configuration block diagram. (**b**) VSI each phase power switch configuration diagram.

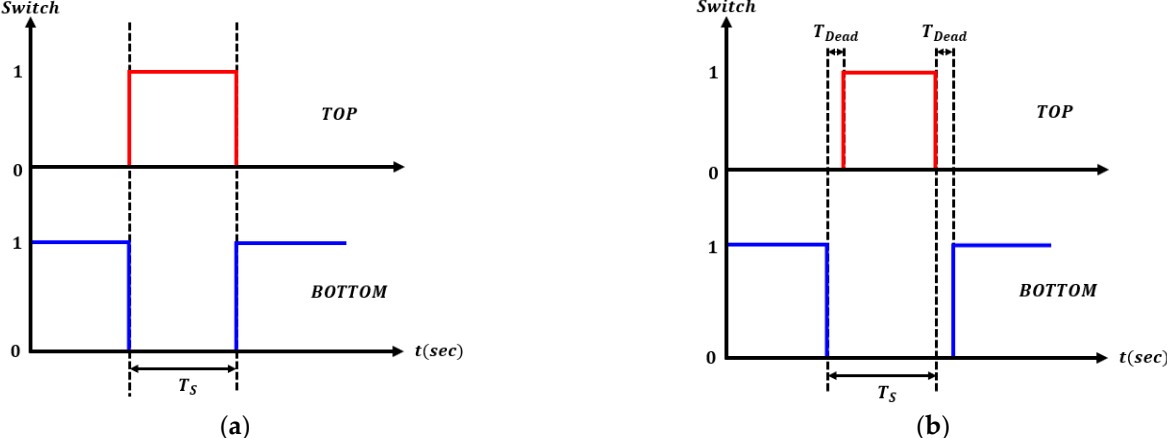

**Figure 6.** (**a**) ON/OFF signal of TOP/BOTTOM power switch element in an ideal case without dead time. (**b**) ON/OFF signal of TOP/BOTTOM power switch element with dead time.

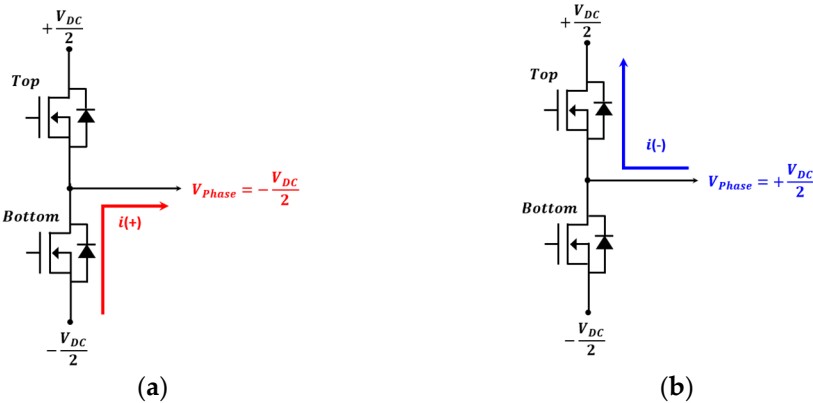

**Figure 7.** (**a**) A situation in which $-V_{DC}/2$ voltage is applied to the motor when the current flows in the positive direction and is in a dead-time condition (**b**) A situation in which $V_{DC}/2$ voltage is applied to the motor when the current flows in the negative direction and is in a dead-time condition.

The disturbance voltage $V_{dead}$ applied to the motor by the dead time can be expressed as Equation (7). As can be seen from Figure 7, the unexpected disturbance voltage is applied as the diode is conducted at the dead time point according to the direction of the three-phase current applied to the motor.

$$V_{dead} = -\frac{T_{dead}}{T_{sample}}(V_{DC}) * sign(i_{phase}) \tag{7}$$

In addition, when the diode conducts in the forward direction during dead time, a diode voltage drop component is added and applied to the motor according to the magnitude of the current. The magnitude of the voltage drop component of the diode and MOSFET changes nonlinearly according to the magnitude of the current. This is shown in Figure 8a,b.

In order to know the exact magnitude of the disturbance voltage, the current value $I_{diode}$ at the point when the diode is conducted must be known exactly. However, in order to accurately measure the magnitude of the phase current of the motor, it must be measured at the point when the voltage vector is applied as a zero vector. With these characteristics, the complex device drive method that senses the current value at the switching frequency, i.e., the highest point or lowest point of the carrier wave, is adopted and used. This is a control method that reflects the characteristics of the DSP (Digital Signal Processor) that schedules tasks according to the CPU (Central Processing Unit) clock. Accurately measuring the current at the point at which the above-mentioned diode is conducting means accurately

reading the ON/OFF status, which is instantaneously variable according to the pole voltage command. However, this measurement method is complicated and vulnerable to noise during sensing, so it is difficult to measure an accurate value.

$$V_{dead\_f} = V_{dead} + V_{diode} \tag{8}$$

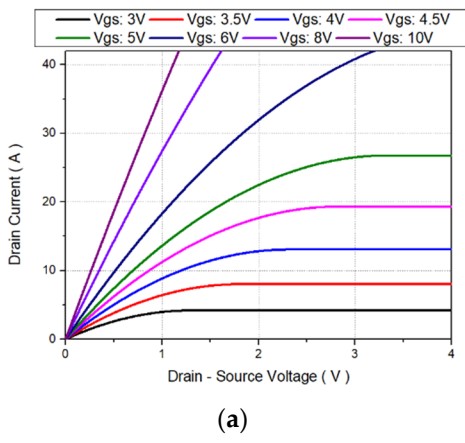 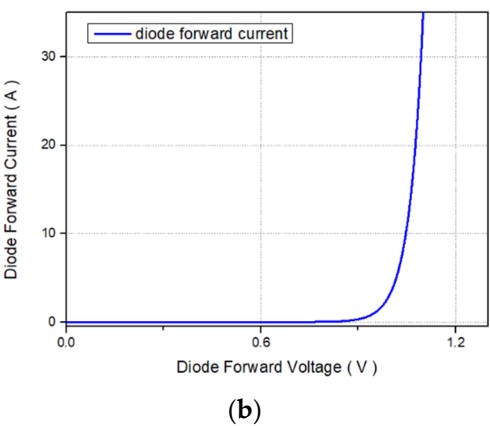

(**a**)        (**b**)

**Figure 8.** (**a**) Turn-on characteristics of the MOSFET (**b**) Turn-on characteristics of the diode.

In this paper, the disturbance voltage generated during dead time is estimated using an observer. The disturbance voltage to be estimated can be expressed as Equation (8), where $V_{dead}$ is the unexpected disturbance pole voltage applied to the motor by each phase current direction, $V_{diode}$ is the voltage loss caused by the power switch elements and diodes, and $V_{dead\_f}$ is the total disturbance voltage during dead time. $V_{dead\_f}$ can be expressed as Equation (9) through the coordinate transformation on the *d*-*q*-axis coordinate plane used for control.

$$
\begin{bmatrix} V_{d\_dead}^r \\ V_{q\_dead}^r \end{bmatrix} = T(\theta_r) \begin{bmatrix} V_{dead\_f} * sign(i_a) \\ V_{dead\_f} * sign(i_b) \\ V_{dead\_f} * sign(i_c) \end{bmatrix}
$$
$$
= \frac{2}{3} \begin{bmatrix} \cos(\theta_r) & \cos(\theta_r - \frac{2\pi}{3}) & \cos(\theta_r + \frac{2\pi}{3}) \\ -\sin(\theta_r) & -\sin(\theta_r - \frac{2\pi}{3}) & -\sin(\theta_r + \frac{2\pi}{3}) \end{bmatrix} \begin{bmatrix} V_{dead\_f} * sign(i_a) \\ V_{dead\_f} * sign(i_b) \\ V_{dead\_f} * sign(i_c) \end{bmatrix} \tag{9}
$$

The disturbance voltage values, $V_{d\_dead}^r$ and $V_{q\_dead}^r$, generated during the dead time and obtained through Equation (9) are shown in Figure 9 on the *d*-*q*-axis coordinate plane through MATLAB R2022a simulation. In addition, the disturbance voltage applied to each phase of the motor at the dead time is expressed in Figure 10b. The ideal gate voltage and the gate voltage considering the dead time and the pole voltage output from VSI to the motor, when the sign of the current output to the phase is negative and positive are shown in Figure 10a. The disturbance voltage causes the distortion of the current waveform applied to the actual three phases of the stator. This can be confirmed in Figure 11. In the case of the DTC algorithm, as was mentioned in the previous section, the stator flux is estimated using the measured three-phase currents. Therefore, the distortion of the three-phase current also appears in the trajectory of the stator flux estimated in the DTC−SVM algorithm. In addition, it can be predicted that the distortion of the estimated stator flux trajectory eventually causes torque ripple during the DTC−SVM algorithm. This was demonstrated through simulations and experiments.

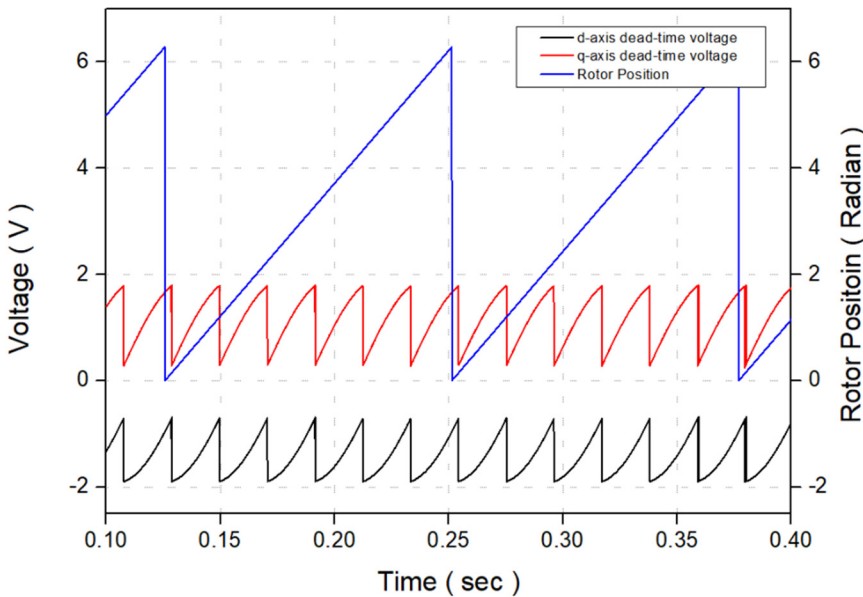

**Figure 9.** Distorted voltage due to dead time appearing on the $d-q$ coordinates waveform $V_{d\_dead}^r$, $V_{q\_dead}^r$, and the motor rotor angle waveform $\theta_r$.

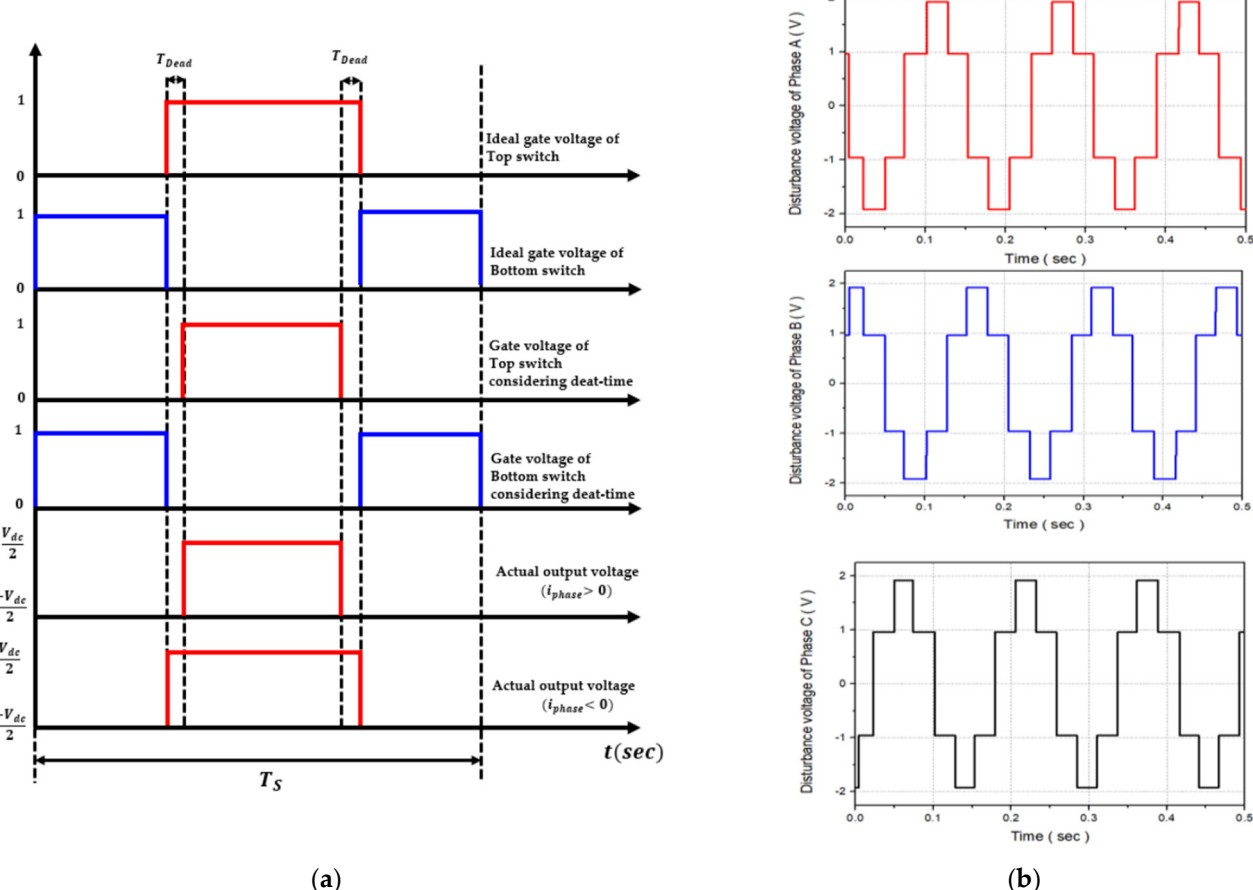

(**a**)   (**b**)

**Figure 10.** (**a**) Influence of dead−time on VSI output voltage, (**b**) Disturbance voltage generated in each phase of the motor by the dead time.

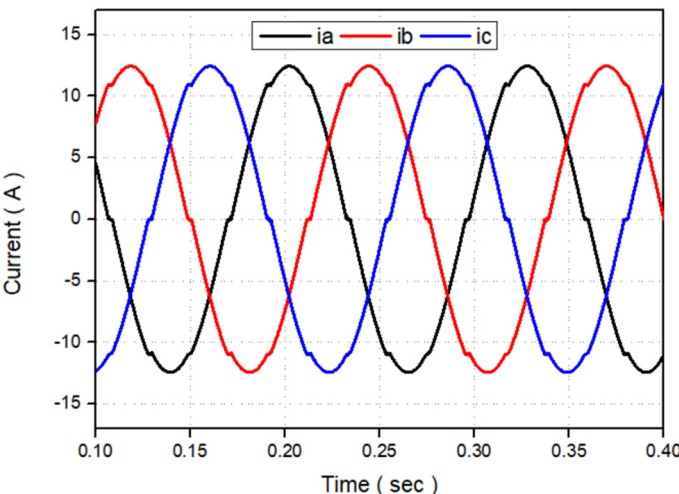

**Figure 11.** Three−phase current waveform with harmonics as the disturbance voltage generated during dead time.

## 4. Proposed Disturbance Voltage Compensation Method

The voltage distortion that occurs during the dead time changes depending on the operating state of the motor. In addition, an additional circuit configuration is required to measure the current and voltage of the inverter; otherwise, it is difficult to measure directly. Therefore, to overcome these problems, this paper proposes a method of compensating the DTC-SVM three-phase control voltage output by designing a disturbance observer based on the PMSM equation of state and motor parameters.

The proposed disturbance observer uses the DC voltage and the three-phase current configured in the VSI without the need to add a hardware circuit. In the previous sections, it was demonstrated that the disturbance voltage generated during the dead time causes distortion of the voltage output from the VSI. It was also confirmed that the distorted voltage also causes distortion in the three-phase current of the motor, resulting in distortion in the stator flux estimated by the controller. The proposed disturbance observer uses the command voltage output from the DTC-SVM controller and the distorted voltage and current values generated by the dead time situation to estimate the distorted voltages $V_{d\_dead}^{r}$ and $V_{q\_dead}^{r}$ generated in the $d$-$q$-axis coordinate system, which rotate in synchronization with the rotor. The distortion voltages $V_{d\_dead}^{r}$ and $V_{q\_dead}^{r}$ on the $d$-$q$-axis coordinate system estimated through the disturbance observer feed the three-phase command voltage output to the controller, thereby compensating for the distortion voltage generated during the dead time. The configuration diagram of the proposed algorithm is shown in Figure 12.

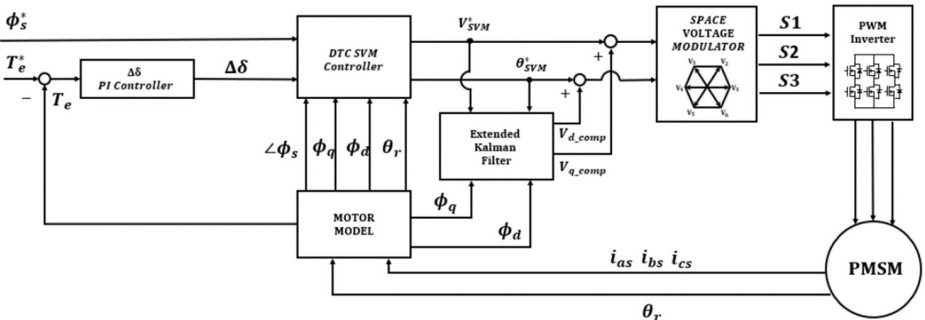

**Figure 12.** DTC-SVM control block diagram applying the proposed extended Kalman filter-based disturbance observer.

In the case of the proposed disturbance observer, the voltage distortion caused by the dead time has nonlinearity; therefore, it was designed using the extended Kalman filter.

Equation (10) shows the distorted voltage value generated by the dead time added to the *d-q*-axis model equation that rotates synchronously with the PMSM rotor.

$$
\begin{bmatrix} V_d^* \\ V_q^* \end{bmatrix} = \begin{bmatrix} V_d + V_{d\_comp} \\ V_q + V_{q\_comp} \end{bmatrix} = \begin{bmatrix} R_s + \frac{d}{dt}L_d & -w_r L_q \\ w_r L_d & R_s + \frac{d}{dt}L_q \end{bmatrix} \begin{bmatrix} i_d \\ i_q \end{bmatrix} + \begin{bmatrix} 0 \\ w_r \phi_f \end{bmatrix} + \begin{bmatrix} V_{d\_dead}^r \\ V_{q\_dead}^r \end{bmatrix}
\tag{10}
$$

$V_d$, $V_q$, $i_d$, and $i_q$ represent the voltage and current values of the *d-q*-axis rotor coordinate system, rotating in synchronization with the rotor. $R_S$, $L_d$, $L_q$, $w_r$, and $\phi_f$ represent the stator resistance, *d-q*-axis inductance values, rotor angular velocity, and rotor flux values, respectively. $V_{d\_comp}$ and $V_{q\_comp}$ represent the voltage compensation value on the *d-q*-axis coordinate system, and $V_{d\_dead}^r$ and $V_{q\_dead}^r$ represent the distortion voltage on the *d-q*-axis coordinate system caused by dead time. The voltage compensation values $V_{d\_comp}$ and $V_{q\_comp}$ on the *d-q*-axis coordinate system are the same as the dead time distortion voltage values. The disturbance observer using the EKF (extended Kalman filter) needs to be modified by a state space model expressed as Equation (11).

$$
\begin{aligned}
x_k &= f(x_{k-1}) + w_{k-1} \\
z_k &= H x_{k-1} + v_{k-1}
\end{aligned}
\tag{11}
$$

In order to modify in the form of the EKF equation of state, the state variable must first be selected. In this paper, the DTC-SVM algorithm is performed, and the *d-q*-axis stator flux and dead time distortion voltages $\phi_d$, $\phi_q$, $V_{d\_dead}^r$, and $V_{q\_dead}^r$ are selected as state variables to compensate for the distorted voltage generated during the dead time. Substituting Equation (2) into Equation (10) to apply the selected state variable to the EKF can be expressed as Equation (12).

$$
\begin{bmatrix} \frac{d}{dt}\phi_d' \\ \frac{d}{dt}\phi_q \end{bmatrix} = \begin{bmatrix} -\frac{R_s}{L_d} & w_r \\ -w_r & -\frac{R_s}{L_q} \end{bmatrix} \begin{bmatrix} \phi_d' \\ \phi_q \end{bmatrix} + \begin{bmatrix} 1 & 0 \\ 0 & 1 \end{bmatrix} \begin{bmatrix} V_d^* \\ V_q^* \end{bmatrix} + \begin{bmatrix} -1 & 0 \\ 0 & -1 \end{bmatrix} \begin{bmatrix} V_{d\_dead} \\ V_{q\_dead} \end{bmatrix} + \begin{bmatrix} 0 \\ w_r \phi_f \end{bmatrix}
\tag{12}
$$

After that, if the state equation is organized into a system model and a measurement model is used in EKF, it can be expressed as Equations (13) and (14). Assuming that the distortion voltage caused by the dead time among the state variables does not change rapidly, it can be expressed as Equation (13).

$$
\begin{aligned}
\begin{bmatrix} \frac{d}{dt}\phi_d' \\ \frac{d}{dt}\phi_q \\ \frac{d}{dt}V_{d\_dead}^r \\ \frac{d}{dt}V_{q\_dead}^r \end{bmatrix} &= \begin{bmatrix} -\frac{R_s}{L_d} & w_r & -1 & 0 \\ -w_r & -\frac{R_s}{L_q} & 0 & -1 \\ 0 & 0 & 0 & 0 \\ 0 & 0 & 0 & 0 \end{bmatrix} \begin{bmatrix} \phi_d' \\ \phi_q \\ V_{d\_dead}^r \\ V_{q\_dead}^r \end{bmatrix} + w \\
&= f(x) + w
\end{aligned}
\tag{13}
$$

$$
\begin{aligned}
\begin{bmatrix} \phi_d' \\ \phi_q \end{bmatrix} &= \begin{bmatrix} 1 & 0 & 0 & 0 \\ 0 & 1 & 0 & 0 \end{bmatrix} \begin{bmatrix} \phi_d' \\ \phi_q \\ V_{d\_dead}^r \\ V_{q\_dead}^r \end{bmatrix} + v \\
&= Hx + v
\end{aligned}
\tag{14}
$$

The observer design was carried out using the system model and measurement model of the EKF organized in this way. Since the corresponding system model is nonlinear, the nonlinear system model was linearized through a linearization method using the Jacobian determinant. The matrix equation organized by the Jacobian matrix is shown in Equation (15).

$$
A = \begin{bmatrix}
\frac{\partial f_1}{\partial \phi_d'} & \frac{\partial f_1}{\partial \phi_q} & \frac{\partial f_1}{\partial V_{d\_dead}^r} & \frac{\partial f_1}{\partial V_{q\_dead}^r} \\
\frac{\partial f_2}{\partial \phi_d'} & \frac{\partial f_2}{\partial \phi_q} & \frac{\partial f_2}{\partial V_{d\_dead}^r} & \frac{\partial f_2}{\partial V_{q\_dead}^r} \\
\frac{\partial f_3}{\partial \phi_d'} & \frac{\partial f_3}{\partial \phi_q} & \frac{\partial f_3}{\partial V_{d\_dead}^r} & \frac{\partial f_3}{\partial V_{q\_dead}^r} \\
\frac{\partial f_4}{\partial \phi_d'} & \frac{\partial f_4}{\partial \phi_q} & \frac{\partial f_4}{\partial V_{d\_dead}^r} & \frac{\partial f_4}{\partial V_{q\_dead}^r}
\end{bmatrix}
\tag{15}
$$

In addition, since the current system is a discretization model, the EKF system model and the Jacobian matrix must be discretized. If the sampling time $T_s$ is sufficiently small, discretization is possible using the Euler integral. Therefore, the system model of this paper was also discretized using the Euler integral and expressed as Equation (16).

$$
\begin{bmatrix} \phi_d'(k) \\ \phi_q(k) \\ V_{d\_dead}^r(k) \\ V_{q\_dead}^r(k) \end{bmatrix} = \begin{bmatrix} -\frac{R_s T_s}{L_d} & T_s w_r & -T_s & 0 \\ -T_s w_r & -\frac{R_s T_s}{L_q} & 0 & -T_s \\ 0 & 0 & 0 & 0 \\ 0 & 0 & 0 & 0 \end{bmatrix} \begin{bmatrix} \phi_d'(k-1) \\ \phi_q(k-1) \\ V_{d\_dead}^r(k-1) \\ V_{q\_dead}^r(k-1) \end{bmatrix} + \begin{bmatrix} \phi_d'(k-1) \\ \phi_q(k-1) \\ V_{d\_dead}^r(k-1) \\ V_{q\_dead}^r(k-1) \end{bmatrix} + Q
$$

$$
= (T_s \tfrac{d}{dt} f(x_{k-1}) + x_{k-1}) + Q
$$

$$
\begin{bmatrix} \phi_d'(k) \\ \phi_q(k) \end{bmatrix} = \begin{bmatrix} 1 & 0 & 0 & 0 \\ 0 & 1 & 0 & 0 \end{bmatrix} \begin{bmatrix} \phi_d'(k) \\ \phi_q(k) \\ V_{d\_dead}^r(k) \\ V_{q\_dead}^r(k) \end{bmatrix} + R
$$

$$
= H x_k + R
$$

$$(16)$$

As an input factor to the EKF algorithm, the *d-q*-axis command voltage values, including compensation voltages $V_{d\_comp}(k-1)$ and $V_{q\_comp}(k-1)$, were configured. It is also assumed that the rotor magnetic flux of the motor is constant. $w_r$ is updated at each step of the algorithm. In addition, matrices $Q$ and $R$ represent the covariance matrix of process noise and the diagonal matrix of covariance of measurement noise, respectively, and were selected as shown in Equation (17), assuming that the noise of the current system is very small. Additionally, the system model matrix A used in the EKF is discretized and shown in Equation (18), where $I$ is a $4 \times 4$ identity matrix.

$$
Q = \begin{bmatrix} 0.1 & 0 & 0 & 0 \\ 0 & 0.1 & 0 & 0 \\ 0 & 0 & 0.1 & 0 \\ 0 & 0 & 0 & 0.1 \end{bmatrix}
$$

$$(17)$$

$$
R = \begin{bmatrix} 0.1 & 0 \\ 0 & 0.1 \end{bmatrix}
$$

$$
A(k) = (I + T_s A) = \begin{bmatrix} 1 - \frac{R_s T_s}{L_d} & w_r T_s & -T_s & 0 \\ -w_r T_s & 1 - \frac{R_s T_s}{L_q} & 0 & -T_s \\ 0 & 0 & 1 & 0 \\ 0 & 0 & 0 & 1 \end{bmatrix}
$$

$$(18)$$

The finally proposed disturbance observer using the EKF is shown in Equations (19)–(23) below.

$$
\overline{x}(k) = f(\overline{x}(k-1))
$$

$$(19)$$

$$
\overline{P}(k) = A\hat{P}(k-1)A^T + Q
$$

$$(20)$$

$$
K = \overline{P}(k)H^T(H\overline{P}(k)H^T + R)^{-1}
$$

$$(21)$$

$$
\hat{x}(k) = \overline{x}(k) + K(z(k) - H\overline{x}(k))
$$

$$(22)$$

$$
\hat{P}(k) = \overline{P}(k) - KH\overline{P}(k)
$$

$$(23)$$

where $K$ is the gain of the Kalman filter and $P$ is the covariance vector of the state variable. Additionally, $\hat{x}(k)$ and $\hat{P}(k)$ denote the estimated state variable and covariance and $\overline{x}(k)$,

$\overline{P}(k)$ denote the predicted value of the state variable and covariance. The covariance vector $P(k)$ is composed of a $4 \times 4$ matrix, and the covariance vector matrix is a diagonal matrix. The initial value of the covariance vector was designed to be 10. The dead time distortion compensation voltage value on the $d$-$q$-axis coordinate system $V_{d\_comp}$ and $V_{q\_comp}$, which is the state variable of the disturbance voltage observer using the designed EKF, is added to the three-phase output voltage command value of the DTC-SVM controller. The values of $V^r_{d\_dead}$, $V^r_{q\_dead}$ and $V_{d\_comp}$, $V_{q\_comp}$ estimated through the designed observer are compared in Figures 13 and 14, respectively. It can be seen from Figures 13 and 14 that the estimated value of the designed disturbance observer follows the actual value well. Based on this, it can be seen that performance was improved by compensating the DTC-SVM controller with the compensation value that was followed using simulation.

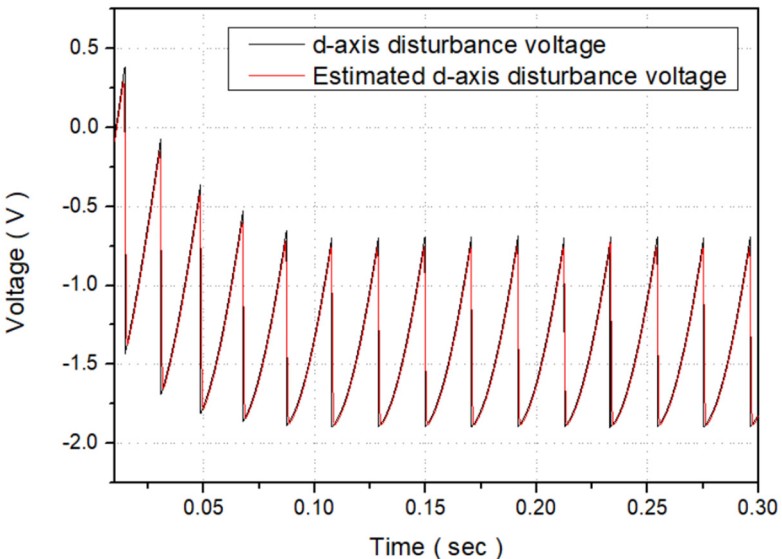

**Figure 13.** The actual disturbance voltage occurring on the $d$-axis during the dead time and the waveform of the $d$-axis disturbance compensation voltage estimated through the disturbance observer.

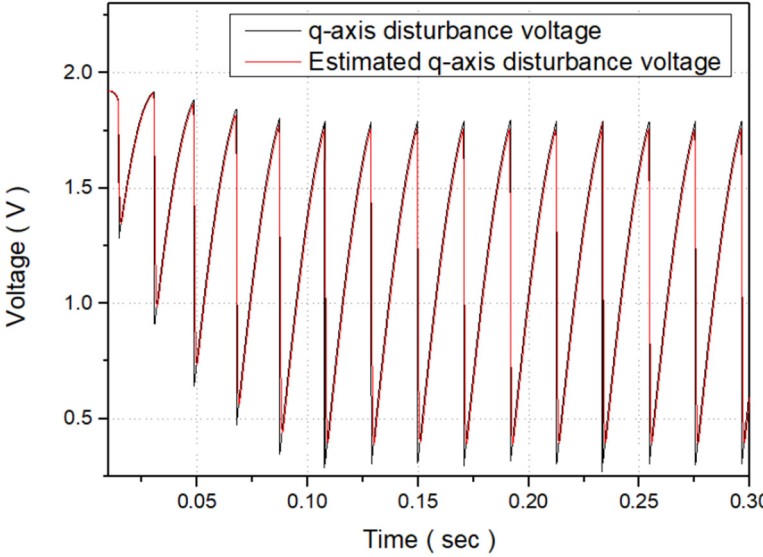

**Figure 14.** The actual disturbance voltage occurring on the $q$-axis during the dead time and the $q$-axis disturbance compensation voltage waveform estimated through the disturbance observer.

## 5. Simulation and Experiment Result of Proposed Algorithm

To validate the performance of the proposed EKF-based disturbance observer, the PMSM operating model in MATLAB/Simulink was used. The simulation was conducted in two environments. The first outputs the stator flux trajectory, torque ripple, and three-phase stator current waveforms in the simulation without the voltage compensation of the disturbance observer. Then, in the simulation, when the voltage compensation of the disturbance observer proposed in this paper is applied, the stator flux trajectory, torque ripple, and phase stator current waveform are output. Afterward, the output results of each of the two simulations were compared in order to assess the improvement in the case of compensation using a disturbance observer. The motor parameters used in the experiments and simulations are shown in Table 2. The experimental setup is shown in Figure 15.

**Table 2.** Motor parameter values.

| Parameter | Value | Unit |
| --- | --- | --- |
| Motor Pole | 8 | |
| DC-Voltage | 48 | V |
| Max. Torque | 2.0 | Nm |
| Phase Resistance | 0.295 | Ω |
| $d$-axis Inductance | 0.22 | mH |
| $q$-axis Inductance | 0.29 | mH |
| Magnetic Flux | 0.0273 | Wb |

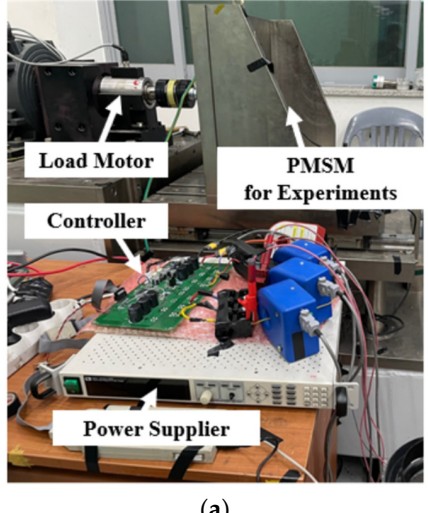

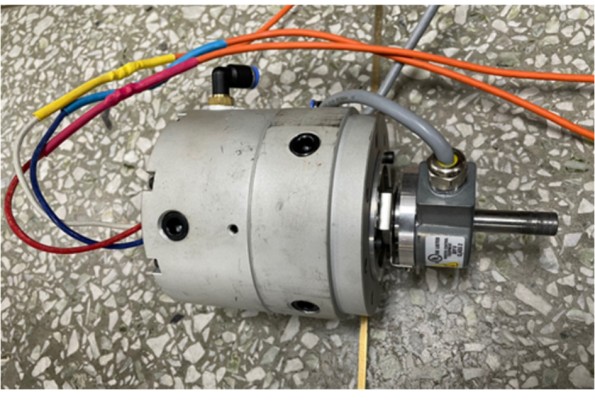

(**a**)  (**b**)

**Figure 15.** (**a**) Experimental setup (**b**) Motors used during the test.

First, Figure 16 shows the stator flux trajectory waveform when controlling a DTC torque of 1.5 Nm. As shown in Figure 16a, it can be seen that the stator flux trajectory is distorted by the distorted voltage generated during the dead time rather than being circular. This confirms that there are six ripples per rotation in the stator flux trajectory, as well as confirming that the disturbance voltage is six times the electrical angular velocity of the motor through the previous section. Figure 16b shows the stator flux trajectory when the voltage is compensated using the proposed disturbance observer. Compared to Figure 16a, it can be confirmed that the ripple seen in the stator flux trajectory is clearly reduced. As mentioned above, the estimated stator flux trajectory has a great influence on the performance of the DTC-SVM controller. Subsequently, Figure 17a,b shows the torque ripple when the voltage is not compensated and when the voltage is compensated by applying the proposed disturbance observer. As shown in Figure 17, it can be confirmed that the torque ripple is significantly reduced. Finally, in Figure 18a,b, the current waveforms of the three phases of the stator are compared with and without the compensation for

the disturbance voltage generated during the dead time. As can be seen through the two waveforms, the distortion of the stator current waveform is significantly reduced.

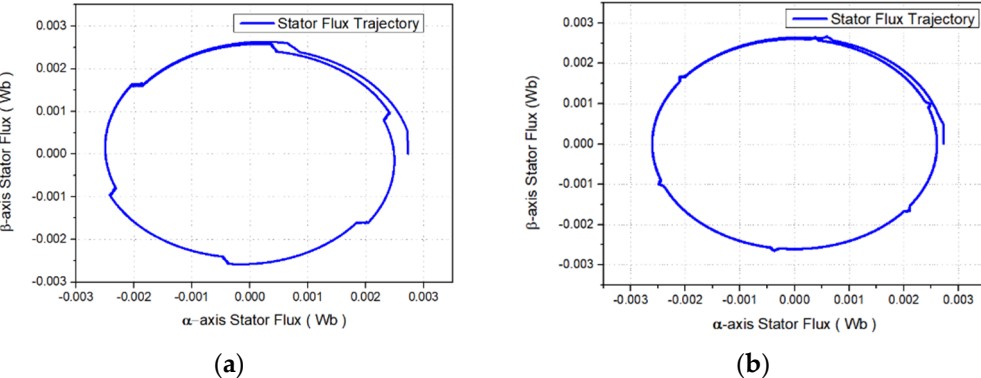

(a)  (b)

**Figure 16.** Trajectory of stator flux during DTC−SVM torque control. (**a**) Scenario with no compensation for disturbance voltage generated during dead time. (**b**) Scenario with disturbance voltage generated during dead time, which is compensated using a disturbance observer.

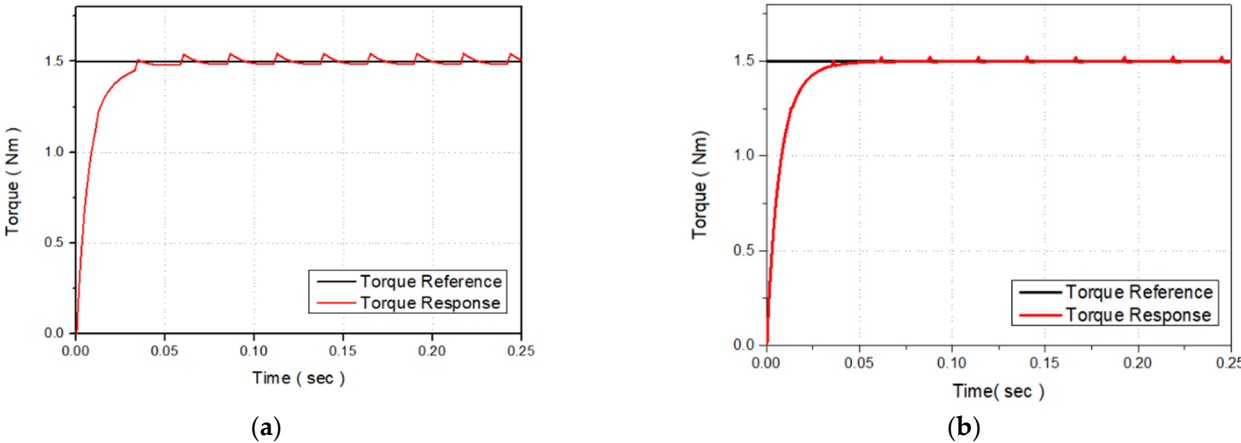

(a)  (b)

**Figure 17.** Torque ripple that occurs during DTC torque control. (**a**) Scenario with no compensation for disturbance voltage generated during the dead time (**b**) Scenario when the disturbance voltage generated during the dead time is compensated using a disturbance observer.

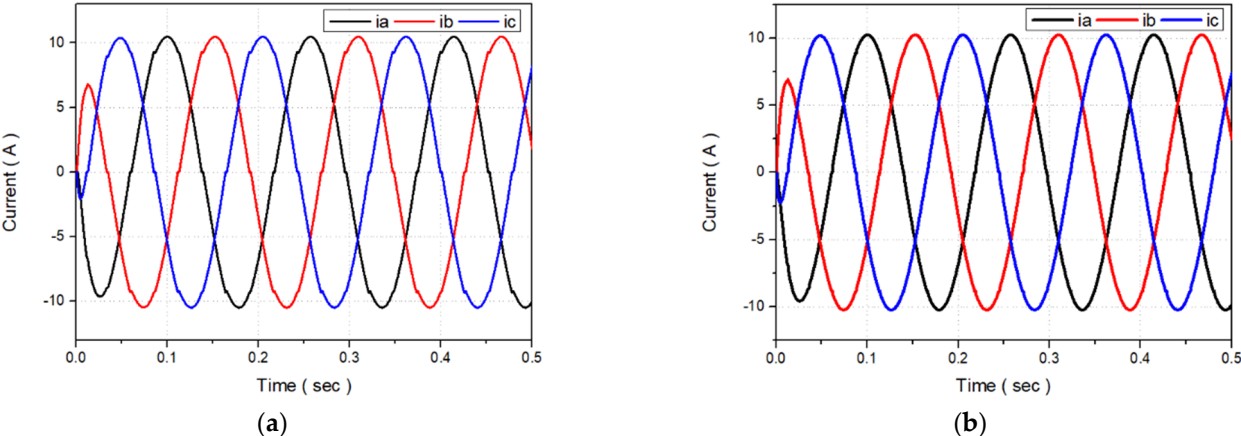

(a)  (b)

**Figure 18.** Three−phase stator current waveform during DTC torque control. (**a**) Scenario with no compensation for disturbance voltage generated during the dead time. (**b**) Scenario when the disturbance voltage generated during the dead time is compensated using a disturbance observer.

In addition, the algorithm based on the extended Kalman Filter proposed in this paper is compared with the conventional compensation algorithm. In the conventional algorithm, the voltage was compensated by considering only the dead time according to the direction of the current applied to each phase of the motor. Therefore, the conventional algorithm could not compensate for torque ripple due to the nonlinearity of voltage drop that varies according to the amount of current through the power switch element and diode. However, the proposed algorithm can compensate for the torque ripple by predicting the nonlinear voltage drop component occurring in the diode and power switch element. A comparison of the average torque ripple is shown in Figure 19a. In addition, The average stator flux ripple when the conventional compensation algorithm is applied and the stator flux ripple when the proposed algorithm is applied are shown in Figure 19b. As shown in Figure 19, it can be seen that the compensation algorithm proposed in this paper is more stable in terms of torque ripple than the conventional algorithm when voltage drop components due to the diode and the power switch device occur. Also in terms of stator flux, the proposed algorithm is more stable than the conventional algorithm. As a result, it was confirmed through the experimental results that the proposed algorithm is more robust to external disturbance than the conventional algorithm.

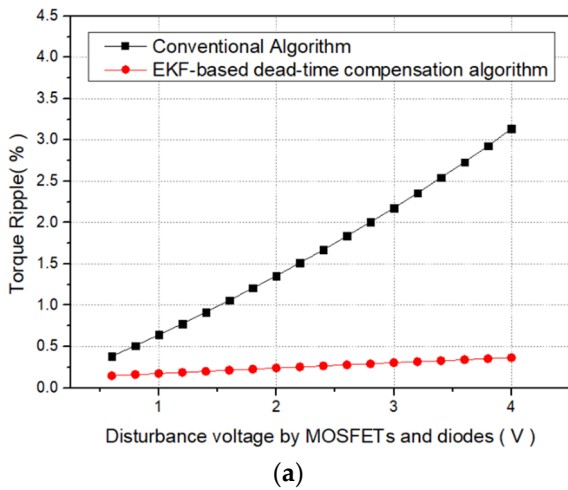 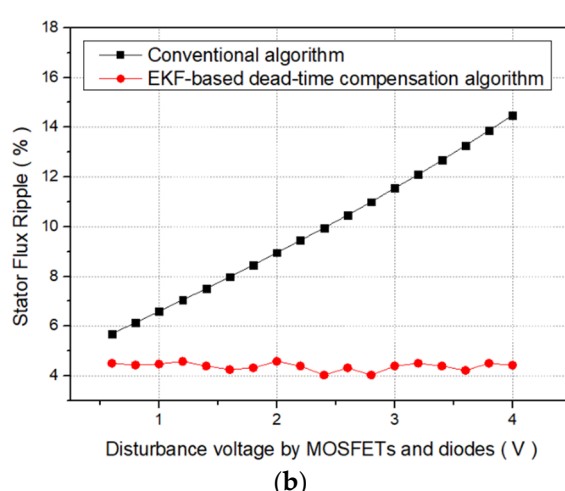

(a)          (b)

**Figure 19.** (**a**) Comparison of torque ripple for disturbance voltage generated by MOSFETs and diodes when applying the conventional algorithm and the proposed EKF-based control algorithm (**b**) Comparison of stator flux ripple for disturbance voltage generated by MOSFETs and diodes when applying the conventional algorithm and the proposed EKF-based control algorithm.

## 6. Conclusions

In this paper, an extended Kalman filter-based disturbance voltage observer is proposed to improve the performance of stator flux estimation and torque ripple induced by the dead time. The performance of the proposed observer was demonstrated through experiments with MATLAB/Simulink. Through the results of the experiment, it was confirmed that the distortion that exists in the trajectory of the estimated stator flux was clearly improved. It was also confirmed that the disturbance observer based on the extended Kalman filter estimated the disturbance voltage very well. In addition, the torque ripple in the conventional algorithm increased by up to 3% of the maximum torque as the disturbance voltage generated by the MOSFET and diode increased. In the case of the algorithm proposed in the paper, the torque ripple is maintained within 0.5%. It was also confirmed that the magnitude of the torque ripple when the proposed algorithm was applied was improved by 20% compared to the torque ripple when the conventional algorithm was applied. In terms of stator flux ripple, when the conventional algorithm was applied, the ripple of the stator flux estimation value increased by up to 15% of the stator flux command due to the disturbance voltage caused by the power switch elements and diodes. Conversely, when the proposed algorithm was applied, the ripple of the stator flux

estimation value was maintained at 5% or less of the stator flux command. Through the experimental results, the proposed algorithm is confirmed to be more stable than the conventional algorithm. Finally, The algorithm proposed in this paper is robust to disturbances in terms of stator flux estimation and the torque ripple, and its performance improves when performing PMSM control based on the DTC-SVM algorithm without using additional hardware circuits or complex measurement methods.

**Author Contributions:** Conceptualization, D.-I.S.; methodology, D.-I.S.; software, D.-I.S.; validation, D.-I.S.; writing—original draft preparation, D.-I.S. and J.-S.H.; writing—review and editing, D.-I.S., J.-S.H., H.-S.L. and J.-S.P.; supervision, G.-H.L. All authors have read and agreed to the published version of the manuscript.

**Funding:** This research was supported by the Korea Evaluation Institute of Industrial Technology (KEIT, Korea) grant funded by the Government of Korea (MOTIE, Korea) (No. 1055001018, Development of 50~150 kW electric drive system for construction machinery).

**Data Availability Statement:** Data sharing is not applicable to this article.

**Conflicts of Interest:** The authors declare no conflict of interest.

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
