# Peer review of "Performance Improvement of DTC-SVM of PMSM with Compensation for the Dead Time Effect and Power Switch Loss Based on Extended Kalman Filter"

_electronics, doi:10.3390/electronics12040966_

Round 1

Reviewer 1 Report

1) The dead-time compensation is a well-known topic, which was perfectly studied during last several decades, however the authors do not mention this fact. The first work in this area relates to 1983.

2) The authors did not review existing dead-time compensation techniques, thus the reader can think that this is the pioneering research, however it is not true.

The list of similar researches:

a) J. Zitzelsberger and W. Hofmann, "Space Vector Modulation with Current Based Dead Time Compensation using Kalman-Filter," IECON 2006 - 32nd Annual Conference on IEEE Industrial Electronics, Paris, France, 2006, pp. 1533-1538, doi: 10.1109/IECON.2006.347612.

b) L. Buchta and O. Bartik, "Dead-Time Compensation Strategies Based on Kalman Filter Algorithm for PMSM Drives," IECON 2019 - 45th Annual Conference of the IEEE Industrial Electronics Society, Lisbon, Portugal, 2019, pp. 986-991, doi: 10.1109/IECON.2019.8926806.

c) L. Buchta and L. Otava, "Adaptive compensation of inverter non-linearities based on the Kalman filter," IECON 2016 - 42nd Annual Conference of the IEEE Industrial Electronics Society, Florence, Italy, 2016, pp. 4301-4306, doi: 10.1109/IECON.2016.7793370.

d) W. Bialkowski, "Application of Kalman filters to the regulation of dead time processes," in IEEE Transactions on Automatic Control, vol. 28, no. 3, pp. 400-406, March 1983, doi: 10.1109/TAC.1983.1103239.

e) D. Diallo, A. Arias and J. Cathelin, "An inverter dead-time feedforward compensation scheme for PMSM sensorless drive operation," 2014 First International Conference on Green Energy ICGE 2014, Sfax, Tunisia, 2014, pp. 296-301, doi: 10.1109/ICGE.2014.6835438.

Besides that there are several minor issues, some of them are pointed out below:

3) The abstract is not sound, it contains a set of weakly connected sentences. It must be rewritten.

4) Denote acronyms before usage.

5) The reviewer does not see the connection between sentences and do not understand the idea “The DTC control technique was widely used in the field of induction motors in their early days. However, with the advantages of high efficiency and high power density, PMSM began to be used in many industrial fields,”

Author Response

Dear reviewer, Thank you again for reviewing the submitted paper.

Attaching a response to the comments written in the review.

Reviewer 2 Report

The paper presents method to improve performance of PMSM drive with compensation of deadtime effect by EKF. The research conducted in the paper is conducted properly however the the results are similar to other papapers (e.g. A Novel Dead-Time Compensation Method using Disturbance Observer), therefore I am curious about the novelty of the paper. In order to prove significance and originality of the paper i would recomend to authors to shorten part about known SVM DTC and add more detailed discussion about deadtime influence and power switch loss analysis; compare their method with others known methods like LUT or othes from the point of view of computational burden, stability, vulnerability to parameter deviation etc and prove it by more graphical results. 

Author Response

Thank you for reviewing this paper.

I provided the responses about the comments in the attached file.

Please see the attachment file.

Reviewer 3 Report

First of all, thank you very much for the material you sent.

For a decade, due to the development of semiconductors and their operating frequency, there has been a return to simple constructions of electrical machines. However, the controls remain very complicated.

The article presents a new approach to control.

However, a change of as much as 20% has not been clearly demonstrated. i.e. whether the change concerns voltage only and in relation to:

- what method,

- what mileage,

- in relation to what.

Please note that many researchers are working on the control of SRM and PMSRM machines. Hence, the state of the art requires a thorough approach.

Best regards. Łukasz Kolimas

Author Response

(The authors gave the same response as above.)

Reviewer 4 Report

This manuscript describes the model by using EKF to improve the performance of DTC-SVM control of PMSM regarding the dead time effect, more specifically the voltage distortion, from VSI. The authors thoroughly go through the modeling and formulation process and demonstrate that the proposed observer shows obvious improvement from Fig.14 to Fig.16. The results presented in this manuscript are interesting while there are some minor revisions needed before publication:

1. By plugging eq.(2) into first formula of eq.(3), it should give us "3/2*P*(phi_f*i_q+(L_d-L_q)*I_d*I_q)" instead of "3/2*P*(phi_f*i_q-(L_d-L_q)*I_d*I_q)". Please double check.

2. In Fig.7, authors proposed to add V_phase to compensate the unwanted current generated from dead time effect. How does it compare to modify the gate voltages of the inverter? What are the advantages of applying phase voltage over adjusting gate voltages for the transistors?

3. How is the covariance matrix of process noise generated with the assumption of sigma^2 to be 0.1 across the board in eq.(18). Would slight change of the values impact the overall Kalman filter result?

Author Response

Thank you for reviewing this paper.

I provided the responses about the comments in the attached file.

Please check the attachment file.

Round 2

Reviewer 2 Report

The revised version of the paper exppains better the idea of authors. The proposed method for deadtime compensation by the means of KF has positive influence on drive performance.